# Atmospheric Thermodynamic Profiling through the Use of a Micro-Pulse Raman Lidar System: Introducing the Compact Raman Lidar MARCO

**DOI:** 10.3390/s23198262

**Published:** 2023-10-06

**Authors:** Paolo Di Girolamo, Noemi Franco, Marco Di Paolantonio, Donato Summa, Davide Dionisi

**Affiliations:** 1Scuola di Ingegneria, Università degli Studi della Basilicata, 85100 Potenza, Italy; noemi.franco@unibas.it (N.F.); marco.dipaolantonio@artov.ismar.cnr.it (M.D.P.); 2Institute of Marine Sciences, National Research Council (ISMAR-CNR), 00133 Roma, Italy; davide.dionisi@artov.ismar.cnr.it; 3Institute of Methodologies for Environmental Analysis (IMAA-CNR), National Research Council, 85050 Tito, Italy; donato.summa@imaa.cnr.it

**Keywords:** water vapor, micro-pulse laser, Raman lidar

## Abstract

It was for a long time believed that lidar systems based on the use of high-repetition micro-pulse lasers could be effectively used to only stimulate atmospheric elastic backscatter echoes, and thus were only exploited in elastic backscatter lidar systems. Their application to stimulate rotational and roto-vibrational Raman echoes, and consequently, their exploitation in atmospheric thermodynamic profiling, was considered not feasible based on the technical specifications possessed by these laser sources until a few years ago. However, recent technological advances in the design and development of micro-pulse lasers, presently achieving high UV average powers (1–5 W) and small divergences (0.3–0.5 mrad), in combination with the use of large aperture telescopes (0.3–0.4 m diameter primary mirrors), allow one to presently develop micro-pulse laser-based Raman lidars capable of measuring the vertical profiles of atmospheric thermodynamic parameters, namely water vapor and temperature, both in the daytime and night-time. This paper is aimed at demonstrating the feasibility of these measurements and at illustrating and discussing the high achievable performance level, with a specific focus on water vapor profile measurements. The technical solutions identified in the design of the lidar system and their technological implementation within the experimental setup of the lidar prototype are also carefully illustrated and discussed.

## 1. Introduction

The feasibility of atmospheric thermodynamic, namely water vapor and temperature, profiling through the application of the Raman lidar technique was demonstrated in the late sixties of the 20th century [1]. Systems exploiting this measurement capability have been developed and operated since then [2,3,4,5,6]. The application of the pure-rotational and roto-vibrational Raman lidar techniques imposes the use of short laser wavelengths, typically in the UV region, as in fact, Raman cross-sections strongly increase in amplitude with a decreasing wavelength, this increase being proportional to the fourth negative power of the wavelength. In the early days, excimer lasers were primarily used as lidar transmitters, these sources being capable to emit UV pulses in the wavelength interval of 308–351 nm, especially suited for this kind of application, e.g., [7]. Excimer lasers are operated at high pulse repetition rates, typically in the range 100–300 Hz, with average UV output powers of 20–50 W. Raman lidars including such laser sources are usually operated within big laboratory infrastructures because of the large sizes of these sources and the careful safety measures which need to be implemented for the secure handling of the gas mixtures used as active media and the operation of the system.

In the early nineties of the 20th century, solid-state Nd:YAG lasers started to be used more and more in Raman lidars. These sources were originally exploiting primarily flash-lamp-based optical pumping. Q-switched flash-lamp-pumped Nd:YAG lasers are typically operated at pulse repetition rates in the range 10–30 Hz, with single-pulse UV energies in the range 100–500 mJ, thus emitting average optical powers in the UV region in the range 1–15 W [8,9,10]. These laser sources are characterized by high peak powers, potentially capable of generating intense thermal shocks and ultimately causing damage within the laser cavity and the transmission chain.

In the last 10 years, in solid-state power laser technology, optical pumping through laser diodes has progressively consolidated as an alternative solution to flashlight pumping, the two technological solutions allowing one to achieve comparable UV average optical powers (10–20 W), but the former doing it based on the consideration of higher pulse repetition frequencies (100–200 Hz) and, consequently, lower single-pulse energies. This translates into smaller peak optical powers hitting the coatings and optical surfaces within the laser cavity and the transmission optics, and consequently, leads to more relaxed damage threshold requirements for the optical and coating surfaces to be used. Eventually, this substantially reduces the risks of hot-spots and damage inside the laser head and in the transmission optics chain. On the other hand, the use of higher pulse repetition frequencies leads to a partial degradation of daytime Raman lidar performances, which may especially affect atmospheric water vapor measurements. In fact, water vapor roto-vibrational signals are the weakest among the signals collected by thermodynamic profiling lidars and the use of higher repetition rates translates into higher amounts of solar background radiation reaching the detectors and consequently higher noise affecting the measurements. This is especially true at the water vapor Raman shifted wavelength at 407.5 nm, located in the peripheral portion of the visible.

An effective alternative to flash-lamp-pumped or diode-pumped Nd:YAG power lasers may be represented by powerful micro-pulse lasers. Micro-pulse lasers are laser sources typically emitting very low energy pulses (hundreds of μJ) with very high repetition rates (1–100 kHz). The most powerful of these sources may emit average optical powers at 355 nm in the range 5–10 W. Several major advantages characterize high-frequency micro-pulse solid state lasers when compared to q-switched flash-lamp or diode-pumped solid-state lasers. Among others: (i) a much lower cost (approximately one order of magnitude smaller), (ii) a much smaller size, and (iii) a longer lifetime. This latter advantage translates into a lower maintenance/refurbishing cost, as in fact, these laser sources are capable of running continuously over 1- or 2-year periods without any major degradation of their performances.

Micro-pulse high-repetition-rate Nd:YAG laser sources have been exploited in lidar science since the late nineties. They have been effectively used to stimulate atmospheric elastic backscatter echoes, and thus for the development of elastic backscatter systems or ceilometers. A variety of micro-pulse backscatter lidars and ceilometers have been developed and deployed all over the world in the last three decades, primarily in the frame of international networks. Among others, the NASA Micro Pulse Lidar Network (MPLNET, [11,12]) is a federated network of micro pulse lidar sites, mostly co-located with the NASA Aerosol Robotic Network (AERONET, [13]), providing information on the vertical properties of both aerosols and clouds. In Europe, the E-PROFILE network (https://e-profile.eu/ (accessed on 28 September 2023)), which is part of the observation programs of the European Meteorological Services Network (EUMETNET), coordinates wind profile measurements from radar wind profilers and attenuated aerosol backscattering profile measurements from automatic lidars and ceilometers (ALC).

Micro-pulse backscatter lidars are particularly effective in measuring atmospheric parameters as the aerosol optical depth and vertical profiles of aerosol backscatter and the depolarization ratio [14,15,16]. They can also provide effective information on cloud boundaries and planetary boundary layer heights, and in the long-term study of urban aerosol properties [17]. These lidar systems are typically compact and lightweight, which makes them easily transportable and suitable for field measurements.

Micro-pulse lasers have also been successfully used in differential absorption lidars (DIALs) to measure water vapor profiles. The DIAL technique represents a very effective and self-calibrating lidar technique to measure atmospheric water vapor. This technique relies on narrowband, high-spectral-fidelity diode lasers, which enable accurate and calibrated measurements, requiring a minimal set of assumptions and a ratio of two signals [18,19]. Narrow-band DIALs with a water vapor profiling capability have been successfully developed in different research institutions [20,21]. The narrow-band DIAL approach relies on laser sources whose spectral width is much narrower than the targeted absorption feature [22]. However, the application of this technique requires the use of complex injection-seeded laser sources capable of guaranteeing high stability and spectral purity. Several alternatives to the narrow-band DIAL approach have been proposed over the years. Among others, the broadband DIAL approach [23]. This relies on very broadband laser sources, spanning hundreds of individual absorption lines, combined with either narrow-band detection [24] or spectrally resolved measurements of the backscatter return [25,26].

A much simpler and cheaper laser source is required for the exploitation of the Raman lidar technique, which is illustrated in this paper. However, the use of micro pulse lasers cannot be easily extended to Raman lidars as in fact, as already mentioned above, the use of high-pulse repetition frequencies typically leads to the sensitive performance degradation of Raman lidars as a result of the high solar background radiation/noise collected when operating lidar systems at high pulse repetition rates. This problem obviously becomes even more important when dealing with micro-pulse lasers, operating at very high repetition rates (1–100 kHz). However, based on the exploitation of the most powerful of these laser sources, in combination with an appropriate compact design of the receiver and a reduced beam divergence, Raman lidar measurements of thermodynamic parameters result to be still feasible with a good signal-to-noise ratio both in the daytime and night-time. A prototype lidar system based on the use of a micro-pulse laser source has been recently developed at Scuola di Ingegneria, Università della Basilicata. The technical solutions identified in the design of the system and their technological implementation within the experimental setup are illustrated and discussed in the present paper. Measurements from this system, with a specific focus on water vapor profiles, are also illustrated and discussed in the paper, where an accurate assessment of measurement performance is also provided. The prototype system, named MARCO (Micro-pulse Atmospheric Optical Radar for Climate and Weather Observations), was successfully deployed in Camargue (Port Saint Louis) and has continuously operated, starting in mid-October 2022, over a period of more than 11 months (up to now) in the frame of the Water vapor Lidar Network Assimilation (WaLiNeAs) project [27].

The layout of the paper is the following. Section 2 illustrates the theory behind the Raman lidar technique applied to water vapor profile measurements and the simulations carried out to assess the expected system performance. Section 3 illustrates the system set up, with a specific focus on the technical solutions identified in the design of the system and their technological implementation. Measurements from the instrument, carried out in the frame of WaLiNeAs, are illustrated in Section 4. Results are summarized in Section 5, where future plans and forthcoming activities are also illustrated.

## 2. Theoretical Considerations and Preliminary Simulations

The range-resolved measurement of the roto-vibrational Raman scattering echoes generated by water vapor, oxygen, and nitrogen molecules when solicited through laser radiation pulses represents a selective and sensitive method for measuring the vertical profiles of the atmospheric water vapor mixing ratio and temperature. As the laser pulses propagate through the atmosphere, part of their energy is elastically backscattered by atmospheric molecules and particles—typically aerosols or hydrometeors—and inelastically backscattered by molecules. Each different molecular species produces echoes with a specific frequency shift.

The Raman lidar techniques, exploiting the pure-rotational and roto-vibrational Raman scattering phenomena, are very powerful lidar techniques as in fact, they allow for measuring a wide range of thermodynamic and compositional properties of the atmosphere based on the use of a single laser wavelength and the implementation of an adequate receiving system, including a number of spectrally separated measuring channels, each one tuned on the specific frequency shift of the considered Raman scattering phenomenon. 

An important limitation of the Raman techniques is represented by the small cross-section of the Raman scattering effect, which is three or more orders of magnitude smaller than the Rayleigh scattering cross-section. Thus, the exploitation of these techniques imposes the use of powerful laser sources and large aperture telescopes and leads to accurate measurements only when focusing on abundant atmospheric molecular species, such as H_2_O, N_2_, O_2_, and CO_2_.

Raman lidar measurements of the water vapor mixing ratio profile have been extensively reported in the literature [28,29,30,31]. The approach relies on the use of the roto-vibrational Raman lidar signals from water vapor, $P_{H_{2}O}\left(z\right)$, and molecular nitrogen, $P_{N_{2}}\left(z\right)$, at the two Raman-shifted wavelengths $\lambda_{H_{2}O}$ and $\lambda_{N_{2}}$, respectively. The vertical profile of the water vapor mixing ratio is obtained through the following equation [32]:(1)$$x_{H_{2}O}\left(z\right)=K\;\cdot \Delta Trs\left(z\right)\cdot \frac{P_{H_{2}O}\left(z\right)}{P_{N_{2}}\left(z\right)}$$
where K is a calibration constant [33,34,35,36,37] and ΔTrs(z) is a differential transmission term, which accounts for the different atmospheric transmissions by molecules and aerosols at $\lambda_{H_{2}O}$ and $\lambda_{N_{2}}$ through the altitude interval from the lidar station level to scattering volume z.

The background-subtracted signals, expressed as the number of detected photons from a given altitude z above station level, are given by the following expression:(2)$$P_{H_{2}O/N_{2}}\left(z\right)=P_{0}\frac{c∆t}{2}\;\frac{A}{z^{2}}\;\eta_{H_{2}O/N_{2}}\;n_{H_{2}O/N_{2}}\left(z\right)\;\sigma_{H_{2}O/N_{2}}\;T_{\lambda }{}_{H_{2}O/N_{2}}\left(z\right)\;T_{\lambda }{}_{0}\left(z\right)$$
where $P_{0}$ is the number of emitted photons for each laser pulse at wavelength $\lambda_{0}$, c is the speed of light, A is the telescope aperture area, $\eta_{H_{2}O/N_{2}}$ is the overall transmitter–receiver efficiency at wavelength $\lambda_{H_{2}O/N_{2}}$, ∆t is the temporal sampling resolution, primarily driven by the detector response time, $n_{H_{2}O/N_{2}}$ represents the water vapor/molecular nitrogen number density, $\sigma_{H_{2}O/N_{2}}$ is the water vapor/molecular nitrogen roto-vibrational Raman cross section, and $T_{\lambda }{}_{0}\left(z\right)$ and $T_{\lambda }{}_{H_{2}O/N_{2}}$ are the atmospheric transmission profiles from station level up to the scattering volume altitude z at $\lambda_{0}$ and $\lambda_{H_{2}O/N_{2}}$, respectively.

The calibration coefficient K in expression (1) may be determined through a calibration procedure based on the comparison between simultaneous and co-located water vapor mixing ratio profiles measured by the lidar and by an independent humidity sensor. For the purpose of this study, the estimate of K is based on an extensive comparison between MARCO and the radiosonde data from a nearby station. The approach is described in detail in [35].

It is to be underlined that signals coming out of the detectors include a signal contribution from the lidar echo and an additional signal contribution accounting for the environmental background radiation reaching the detector. While during night-time lidar operation this latter contribution is typically negligible, in daytime, as a result of the high solar irradiance entering the telescope’s field of view, this background contribution may become comparable or even larger than the lidar echo intensity, this being associated with a large increase of the statistical uncertainty affecting the measurements. This statistical uncertainty can be determined in a variety of ways. One of the major advantages of the lidar techniques is represented by their capability to provide profiles of atmospheric parameters with accurate information on their statistical measurement uncertainty. Statistical uncertainty is usually estimated through the application of Poisson statistics, which is well suited in case of data acquired in the photon counting mode. Whereas signal photon counts are directly measured by the photon counting unit, “virtual” counts can also be obtained from the signals measured by the analogue module (ref. [38], which allows Poisson statistics to be effectively applied also to analogue signals.

In order to get an estimate of the error affecting the water vapor mixing ratio and temperature measurements through Poisson statistics, it is necessary to first apply Poisson statistics to the photon counts of the individual lidar signals contributing to the measurements and then, through error propagation, compute the overall error affecting the measured atmospheric variables. The percentage statistical error affecting the water vapor mixing ratio measurements can be determined based on the application of Poisson statistics through the following analytical expression [32]:(3)$$\frac{\Delta \chi_{H_{2}O\;}\left(z\right)}{\chi_{H_{2}O}\left(z\right)}=100\;\sqrt{\frac{P_{H_{2}O}\left(z\right)+bk_{H_{2}O}}{P_{H_{2}O}^{2}\left(z\right)}+\frac{P_{N_{2}}\left(z\right)+bk_{N_{2}}}{P_{N_{2}}^{2}\left(z\right)}}$$
where the terms $bk_{H_{2}O}$ and $bk_{N_{2}}$ represent the sky background signal collected in the water vapor and molecular nitrogen channels, respectively.

$\Delta \chi_{H_{2}O\;}\left(z\right)/\chi_{H_{2}O\;}\left(z\right)$ includes two contributions, one associated with the random uncertainty affecting $P_{H_{2}O}\left(z\right)$ and one associated with the random uncertainty $P_{N_{2}}\left(z\right)$, with the uncertainty affecting $P_{H_{2}O}\left(z\right)$ being typically larger than the one affecting $P_{N_{2}}\left(z\right)$.

## 3. Instrumental Setup

The present section illustrates the different technological solutions implemented in the development of the compact Raman lidar setup and defines the technical specifications of the different subsystems therein. The short name of the developed system is MARCO, which is an acronym of the extended name “Micro-pulse Atmospheric Optical Radar for Climate and Weather Observations”. The name is given in remembrance and commemoration of Prof. Marco Cacciani, a brilliant lidar scientist and a mentor for a large portion of the Italian lidar community, who passed away in January 2022, as a recognition for all his dedicated effort and service to the community. 

The lidar transmitter has been developed around a high-power UV micro-pulse laser. The selected micro-pulse source was developed by Sintec Optronics Pte. Ltd., Singapore (model: STC-AO-V-355-Water). The laser is: (i) all solid state, (ii) diode-pumped, and (iii) Q-switched, based on acousto-optic modulation (AOM). It operates at 355 nm and features a high peak power, high repetition rate, and short pulse duration. The technical specifications of the laser source are listed in Table 1. The pulse repetition rate may be changed within the interval 10–40 kHz, with different single pulse energies at the different frequencies (Table 1). During the measurements reported in this paper, the laser was operated at a nominal frequency of 20 kHz, with a single pulse energy of 250 μJ and an average emitted power of 5 W at 355 nm. This was considered to be the best possible compromise between the system performance and its capability to achieve the scientific goals of the measurement campaign, as in fact, a repetition rate of 30 kHz could have been used, leading to a larger emitted power (6 W), but with ambiguities, associated with the superimposition of lidar echoes from consecutive laser pulses, above 5 km.

It is to be underlined that the used UV micro-pulse laser operates in eye-safety conditions only from a certain distance. UV eye-safety threshold, computed in compliance with the International Standard IEC 60825-1:2014 [39], for exposure times up to 10 s, is quantified as the maximum permissible exposure (MPE) through the empirical formula MPE = 5.6 × 10^3^ × t^0.25^ J m^−2^. For an exposure time of 10 s eye-safety is achieved from an altitude of approximately 240 m. Shorter exposure times lead to lower nominal ocular hazard distances (NOHD), e.g., NOHD_1s_ = 85 m, NOHD_0.1s_ = 20 m, and the system, although not eye-safe at the exit aperture, should be considered safe for aviation. Eye-safety at all distances, including the exit aperture, could be achieved through further expansion of the laser beam.

The selected laser source is very compact, as in fact the external dimensions of the laser head are 450 (L) × 180 (W) × 95 (H) mm, with an overall volume of 6.3 L, comparable to the volume of a shoe box, and the overall weight is 10 kg. The power supply is even smaller, with external dimensions of 200 (L) × 100 (W) × 50 (H) mm (overall volume of 1 L) and an overall weight of 0.9 kg. The total laser power consumption is approx. 300 W (12 V × 25 A). The laser head is water cooled through the use of an external chilling unit, with external dimensions of 580 (L) × 290 (W) × 470 (H) mm, an overall volume of 79 L and an overall weight of 23 kg. The total power consumption of the laser chiller is approx. 460 W (220V × 2.1 A).

The receiver has been built around a Ritchey–Chrétien telescope (company: Omegon.eu, NIMAX GmbH, Landsberg am Lech, Germany, model: 53815-406/3250), including a circular primary mirror (diameter: 406 mm), a secondary mirror (obstruction diameter: 190 mm), and a combined focal length of 3250 mm. The mechanical structure of the telescope is a carbon-fiber truss-tube. The technical specifications of the telescope are listed in Table 2, together with the specifications of the major receiver sub-systems. The collected radiation is divided into different portions entering the different receiving channels through the use of dichroic mirrors and/or beam-splitter plates. In its present configuration, the receiver includes nine separate receiving channels, but only five of these were actually operational during the WaLiNeAS field campaign. A block diagram of the optical layout of the system is illustrated in Figure 1. The five presently operational channels are: (1) the total elastic backscatter signal at 354.7 nm, $P_{355}\left(z\right)$; (2) the elastic backscatter signal with parallel polarization at 354.7 nm, $P_{355}^{\|}\left(z\right)$; (3) the elastic backscatter signal with perpendicular polarization at 354.7 nm, $P_{355}^{┴}\left(z\right)$; (4) the roto-vibrational Raman backscatter signal by N_2_ molecules at 386.7 nm, $P_{N_{2}}\left(z\right)$; and (5) the roto-vibrational Raman backscatter signal by H_2_O at 407.5 nm, $P_{H_{2}O}\left(z\right)$. The four not yet operational additional channels are: (6) the roto-vibrational Raman backscatter signal by CO_2_ at 371.7 nm, $P_{CO_{2}}\left(z\right)$; (7) the pure-rotational Raman backscatter signal by N_2_ and O_2_ molecules at low rotational quantum numbers (at 354.3 nm), $P_{loJ}\left(z\right)$; (8) the pure-rotational Raman backscatter signal by N_2_ and O_2_ molecules at high rotational quantum numbers (at 352.9 nm), $P_{hiJ}\left(z\right)$; and (9) the fluorescence backscatter signal at 450–460 nm, $P_{FL}\left(z\right)$.

Interference filters (IFs) are used in the receiver as spectral selection devices. Considered IFs are characterized by high transmissions at the center wavelengths (T > 80%), narrow transmission bands (FWHM: 0.1–0.3 nm), and high out-of-band rejection (10^-6^-10^-9^). This latter requirement has to be especially verified at the laser emission wavelength, i.e., at 354.7 nm. Specifications of all interference filters are listed in Table 2. IFs with these technical specifications were purchased from Alluxa Inc., Santa Rosa, CA, USA. Signal detection is carried out based on the use of miniaturized photo-detection modules (Hamamatsu Photonics K.K., Shizuoka, Japan, model: H10721-210), including high-gain (2–4 × 10^6^), high-UV quantum efficiency (> 40%) photomultipliers. These photo-detection modules have very small dimensions (22 × 22 × 50 mm) and weight (80 g).

Data acquisition is carried out based on the use of transient recorders, with both an analog and digital sampling capability. In this regard, it is to pointed out that roto-vibrational Raman lidar signals are very weak and, during the daytime portion of the day, they may become substantially smaller than those produced by the solar background radiation. At night, when solar radiation is absent and environmental background is usually limited, weak lidar signals can be effectively sampled in the photo-counting mode. Conversely, in the daytime, especially when the sun is high in the sky (sun zenith angles (SZAs) smaller than 30°), photo-counting results to be ineffective as photons from Raman echoes add up with those produced by solar radiation, with the overall signal by far exceeding the count rate suited for photon counting exploitation. In this case, analog acquisition, carried out through the use of analog-to-digital converters, is to be preferred as, in fact, it allows a direct conversion of the signal photoelectron current into a sampled digital signal. Based on the above considerations, it follows that the appropriate sampling of Raman signals in a 24/7 continuously operational lidar system is only effectively possible through the simultaneous acquisition of each collected lidar signal in both a photon counting and analog–to–digital conversion mode. In the present system, we make use of transient recorders developed by Licel GmbH, Berlin, Germany (model: TR40-16bit-3U). These units combine, in a single acquisition system, both the analog detection of the photo-electron current exiting the photomultipliers and photon counting of the single photo-electrons. This solution allows one to cover a large signal dynamic range (five-to-six orders of magnitude variability). These units feature high temporal resolution in combination with fast signal repetition rates as in fact, they include a powerful A/D converter (16 Bit at 40 MHz) and an 800 MHz fast photon counting system.

All lidar subsystems are included on a compact metal frame, with a parallelepiped shape (Figure 2a). The dimensions of the frame are approximately 1000 (L) × 500 (W) × 1200 (H) mm. The power consumption of the different subsystems (laser head: 300 W, laser chiller: 460 W, detectors, acquisition system and controlling computer: 100 W) results in an overall power supply requirement not exceeding 900 W.

The lidar is hosted in a cabinet conceived for operation in all weather conditions in outdoor environments (Figure 2b). The cabinet is protected against the ingress of dust and high-pressure water jets (protection level IP56). Furthermore, the indoor temperature may be kept constant within ±1 °C, regardless of the external temperature, with this stable behavior being maintained for large outdoor temperature variability in the range −10/+40 °C. During the WaLiNeAs field campaign, the internal temperature was set to a value of 23 °C, with typically observed variability throughout the day in the range 22–26 °C. Relative humidity is also kept constant within the cabinet, with a variability in the range 40–60%. These stable performances were obtained through the use of an air conditioning system based on the inverter technology (company: Rittal, Herborn, Germany, model: SK3185830).

The air conditioning system also needs to be sized to allow an effective removal of the excess heat produced inside the cabinet during lidar operation. The main source of heat is represented by the laser head, which is sucking an electrical power of approximately 300 W and emitting an average UV optical power of 5 W (wall-plug efficiency: 1.66%). Additional heat sources are represented by the laser chiller and the electronics, with an overall estimated head removal requirement of approximately 500 W. The selected air conditioning system has a specified heat removal capacity of 1600 W, which is then well exceeding the effective heat production.

The cabinet can be completely dismantled in order to make the lidar frame easily accessible and allow its insertion/extraction from the cabinet. The external covering of the cabinet is developed in fiberglass (Figure 3). The roof has an inclination of ~3° to favor rain drain and run off. The development of the scientific cabinet required a dedicated “ad hoc” development, which was commissioned to INTERTEC-Hess GmbH, Neustadt/Donau, Germany.

A large aperture quartz window (fused silica, Corning 7980), with an anti-reflection coating on both sides, has been installed on the roof of the cabinet to allow the laser beam and the telescope to access the sky in all-weather conditions. The quartz window, developed by Precision Glass and Optics, Santa Ana, CA, USA, has a circular shape, with a diameter of 500 mm and a thickness of 38.1 mm. The following additional technical specifications were defined for the window: an anti-reflection coating with transmission exceeding 99% in the spectral range 350–410 nm, considering an AOI equal to 0°; polishing is <1/10 wave, surface roughness is smaller than 5 nm, and parallelism <2 arcminutes. The quartz window is watertight sealed on the roof of the cabinet.

The dimensions of the cabinet are approximately 1350 (L) × 1200 (W) × 1730 (H) mm, corresponding to a volume of 2.8 m^3^ (Figure 3). W is 1510 mm in the portion of cabinet’s wall where the air conditioning system is located. The overall weight of the cabinet inclusive of the compact metal frame hosting the lidar is ~500 kg. Such volume and weight values are approximately half of those typically characterizing the most compact transportable Raman lidars for thermodynamic profiling, including diode-pumped or flash lamp-pumped Nd:YAG laser sources. This volume and weight reduction translates into the possibility to sensitively simplify the logistics and costs associated with the transportation and deployment of the lidar system.

Here follow a few signal measurement examples to illustrate the typical system performance and data quality. Figure 4 illustrates the signals, expressed as mean signal photon counts, integrated over a temporal interval of 30 min and applying a running vertical average of 100 m. These measurements were carried out on 29 October 2022, at night (00:00 UTC) and in the central portion of the day (12:00 UTC). The water vapor roto-vibrational Raman signal, $P_{H_{2}O}\left(z\right)$, is found to extend up to an altitude of 2–2.5 km during the day and up to 5 km or higher at night, while the N_2_ roto-vibrational Raman signal, $P_{N_{2}}\left(z\right)$, is found to extend up to an altitude of 3.5–4 km during the day and up to 5 km or higher at night. Concerning the signals $P_{355}\left(z\right)$ and $P_{355}^{\|}\left(z\right)$, these are found to extend up to 5 km or higher for both the day and night, while $P_{355}^{┴}\left(z\right)$ extends up to 3–3.5 km in the daytime and up to 5 km or higher during the night. It is to be pointed out that the data above 5 km are not illustrated because the sampling of the lidar signals is only up to 5.5 km. This results from the fact that the exploitation of a laser source with a 20 kHz repetition rate prevents from using lidar echoes from altitudes above 7.5 km as, in fact, these echoes superimpose to those generated by the previously emitted pulses. So, unambiguous lidar echoes are possible only below 7.5 km, with this upper altitude limit becoming ~5.5 km when considering the intrinsic limitations of the sampling system when dealing with such a high number of lidar echoes. Obviously, the use of micro-pulse lasers with a lower repetition rate would allow one to increase the vertically sampled atmospheric interval.

Figure 4 also includes the corresponding simulated signals, where available, determined through the end-to-end performance model developed at Scuola di Ingegneria, Università della Basilicata. The simulations were obtained by considering, as the input instrumental parameters, the technical specifications defined in this section for the different lidar subsystems. The atmospheric parameters used as input for the simulations are taken from the simultaneous and almost co-located data from a radiosonde launched from the nearby station of Nimes. Aerosol optical properties are simulated based on the use of the median aerosol extinction data from the ESA Aerosol Reference Model of the Atmosphere [40]. Deviation between the measured and simulated signals are found to be in the range 15–20% within the altitude interval 1–4 km. Higher deviations below 1 km are to be primarily associated with the used aerosol model, which tends to overestimate the actual aerosol content and optical loading. The high agreement between the measured and simulated signals testifies that the system is performing as expected, with a high level of optimization and a good coupling between the telescope and the receiving channels.

Figure 5 illustrates the random error profiles affecting these signals obtained through the application of Poisson statistics to signal photon counts. More specifically, panel a illustrates the night-time error profiles, while panel b illustrates the daytime profiles. At night, the signal $P_{H_{2}O}\left(z\right)$ is found to be affected by a random uncertainty smaller than 50% up to 2.8 km and smaller than 100% up to 3.6 km, while it is smaller than 100% up to 1.8 km in the daytime. The signal $P_{N_{2}}\left(z\right)$ is affected by an uncertainty not exceeding 30% up to 3.1 km and smaller than 100% up to 4.4 km at night, while this is smaller than 30% up to 1.8 km and smaller than 100% up to 2.6 km in daytime. The random uncertainty affecting the signals $P_{355}\left(z\right)$, $P_{355}^{\|}\left(z\right)$, and $P_{355}^{┴}\left(z\right)$ at night is smaller than 5% up to 4.5, 4.0, and 2.0 km, respectively, while in the daytime, it is smaller than 20% up to 4.4, 4.3, and 0.7 km, respectively. Based on expression (3), the random error affecting the water vapor mixing ratio measurements at 1.5 km is 21% and 70% in the night-time and daytime, respectively, while at 2.5 km, they are is 47% and >100% in the night-time and daytime, respectively.

## 4. Results

The system MARCO was deployed in Port-Saint-Louis-du-Rhône (France, Lat.: 43.39 N, Lon.: 4.81 E, Elev.: 5 m, Figure 6) and started continuous operation on 19 October 2022 and is still operational (more than 11 months of continuous operation). Over this long measurement period, no more than 30 h of data are missing, which are a result of brief temporary stops for system maintenance or failures. A picture of the measurement site is illustrated in Figure 7, with the lidar on the right side and the van used to transport it on the left side. The calibration of the system was carried out in Potenza, before its transportation to France, based on the use of three co-located radiosondes launched from the nearby IMAA-CNR radiosonde launching facility, located 7 km W-SW of the lidar site in Potenza. No radiosonde launching facility was present in the proximity of Port-Saint-Louis-du-Rhône, with the closest one being Nimes (60 km N-W of the lidar site). Due to the missing radiosondes on-site, the verification of the calibration was carried out based on the consideration of radiosondes launched from Nimes. For this purpose, we consider the radiosonde profiles for those days and times when back-trajectory analyses were revealing the presence of the same mass overpassing the lidar and the radiosonde station at different times. An example of the application of this verification procedure is illustrated in Figure 8a, which illustrates the vertical profiles of the water vapor mixing ratio measured by MARCO, averaged over the 30 min time interval 21:00–21:30 UTC on 28 October 2022, and the corresponding profile from the radiosonde launched from Nimes at 00:00 UTC on 29 October 2022. Figure 8b illustrates the back-trajectory analysis started in Nimes at 00:00 UTC on 29 October 2022. The back-trajectory analysis considers the air masses at altitudes of 500, 1000, and 1500 m. This analysis clearly reveals that air masses reaching Nimes at these altitudes overpassed the lidar site almost 3 h earlier, when the lidar measurements were carried out. The agreement between the two water vapor mixing ratio profiles is quite striking, which supports the hypothesis that the sounded air masses transited over the lidar site before reaching Nimes.

Figure 9a illustrates the time–height cross section of the evolution of the water vapor mixing ratio over the 12-day period from 11:30 on 22 October 2022 until 15:00 on 03 November 2022. The figure is obtained as a sequence of 587 consecutive profiles, each integrated over 30 min and with a vertical resolution of 100 m. Measurements were carried out in all weather conditions and a variety of precipitation events were captured. For example, a precipitation event, with an intensity of 2–3 mm per hour, took place in the early morning on 1 November 2022, with Figure 10 illustrating the radar reflectivity in the area at 06:00 and 08:15 UTC on this same day. Radar reflectivity values are found to be in the range 30–50 dBZ. Few dry intrusions from the stratosphere or upper troposphere are visible on different days (for example, on 24 and 30 October 2022). Water vapor variability throughout the day, associated with convective activity, is visible in a number of consecutive days until the end of October. For all precipitation events, we were also able to measure the associated cloud and precipitation fields, based on the elastic backscatter measurements. For this purpose, we computed the range-corrected elastic signal at 355 nm, RCS355(z), with this quantity well representing the presence of suspended or precipitation particles in the form of aerosol, clouds, and precipitating hydrometeors. Figure 9b illustrates the variability of RCS355(z) over the 12-day period from 11:30 on 22 October 2022 until 15:00 on 03 November 2022. The figure well reveals the precipitating hydrometeors associated with the precipitation event in the early morning on 1 November 2022, already illustrated above, as well as another rain event taking place shortly afterwards on 3 November 2022. 

Performances of the system have been found to be stable over long observational periods. Figure 11 illustrates the time–height cross section of the evolution of the water vapor mixing ratio over four consecutive months (December 2022 through March 2023). In the figure, each panel, representative of a month period, is obtained as a sequence of 1488 consecutive profiles (1344 in February), each integrated over 30 min. A sequence of diurnal and nocturnal daily cycles is visible in the different panels.

The different panels of the figure clearly reveal the large variability of the water vapor content throughout the winter period, with still reasonable large mixing ratio values (up to 10 g kg^−1^ within the atmospheric boundary layer) in December 2022 and early January 2023. This initial period was characterized by relatively high temperatures and a humid environment. Dry conditions are found to characterize the remining part of the month of January and the first half of February, with values not exceeding 5 g kg^−1^. Relatively high water vapor mixing ratio values are found in the time period 14–25 February 2023, with values throughout the boundary layer up to 10 g kg^−1^, and again in the periods 08–14, 17–20, and 23–26 March 2023. A variety of dry intrusion events from the stratosphere or upper troposphere are visible throughout the period (5–6 and 11–12 December 2022, 14–15 January 2023, and others). Relatively high water vapor mixing ratio values are found in the time period 9–14 and 19–24 March 2023, with boundary layer water vapor mixing ratio values up to 10 g kg^−1^. On 11, 13, and 14 March 2023, large values are observed up to ~4 km.

## 5. Summary, Final Considerations, and Future Plans

In this paper we successfully demonstrated the capability of atmospheric thermodynamic profiling by Raman lidar using a micro-pulse laser source, with a specific focus on water vapor profile measurements. In the paper, we illustrate the technical solutions identified in the design of the lidar system and their technological implementation within the experimental setup and we discuss the high achievable performance level. Reported measurements were carried out during the international field campaign WaLiNeAs, which represented a testbed for the qualification, verification, and calibration of this new compact Raman lidar system.

This demonstrated capability opens new opportunities for future thermodynamic profiling Raman lidar networks because of the relatively low cost of the systems, their limited size and weight, and consequently, their ease of transportation in the field. In fact, micro-pulse lasers usually have very small sizes, typically 300–400 (L) × 150–200 (W) × 80–150 (W) mm. The possibility to rely on very compact laser sources ultimately allows one to develop very compact lidar systems with sizes smaller than those typically characterizing the lidars developed around flash-lamp-pumped or diode-pumped lasers. The smallest compact Raman lidars presently available are the system ARTHUS, from the University of Hohenheim, and the system CONCERNING, from the University of Basilicata. Both these systems have sizes of the order of 2 m in height × 2 m in length and 1–1.5 m in width, for an overall volume of 4–6 m^3^, with weights in excess of 1 ton. The system described in the present paper has a volume and a weight which are approximately half of those (2.8 m^3^ and ~500 kg, respectively). A large portion of the overall lidar weight is attributable to the cabinet and, consequently, a factor of two reduction in the volume translates into a weight reduction by the same amount. A “3 m^3^ volume—500 kg weight” system may be easily transported with commercial vehicles or vans not requiring advanced driving certification, and it is thus transportable by individuals owning a basic driving license. This makes the transportation substantially cheaper and logistically easier than one required for the larger systems.

In the present paper we demonstrated the water vapor measurement capability, while temperature measurements were not carried out because, at the time when the system was deployed in Camargue in the frame of WaLiNeAs, the spectral selection devices needed for these measurements were not available yet. This measurement capability will be implemented and tested in a forthcoming paper. Due to the much larger amplitude of the pure-rotational Raman signals exploited for temperature profiling, we are very confident that these measurements will be successfully carried out and this measurement capability properly demonstrated. Preliminary simulations performed with our end-to-end simulator indicate that, considering an integration time of 30 min and a vertical resolution of 100 m, the random uncertainty affecting temperature measurements carried out by the micro-pulse Raman lidar should not exceed 1 K up to an altitude of 4 km and 1 km in the night-time and daytime, respectively.

## Figures and Tables

**Figure 1 sensors-23-08262-f001:**
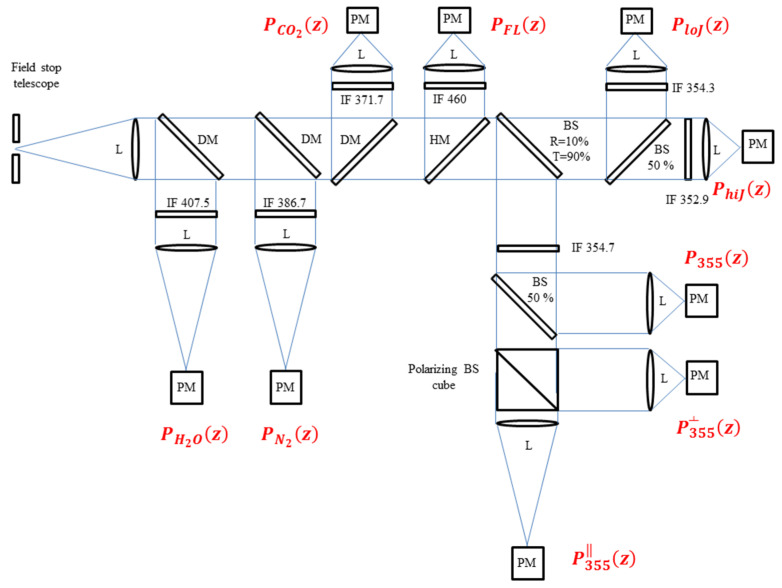
Block diagram of the micro pulse lidar setup. PM: photo-detection modules including photomultipliers, L: lens or refractive collimating system, DM: dichroic mirror, HM: hot mirror, IF: interference filter.

**Figure 2 sensors-23-08262-f002:**
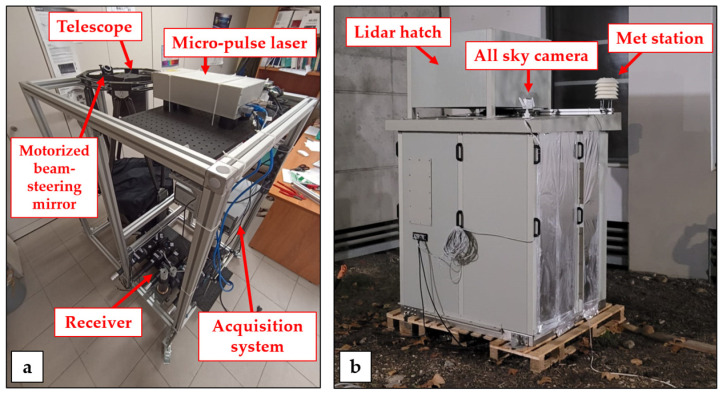
(**a**) Internal view of the Raman lidar system MARCO. The different major sub-systems are clearly visible in the picture (micro-pule laser, telescope, receiver, acquisition system, beam-steering mirror). (**b**) External view of the Raman lidar.

**Figure 3 sensors-23-08262-f003:**
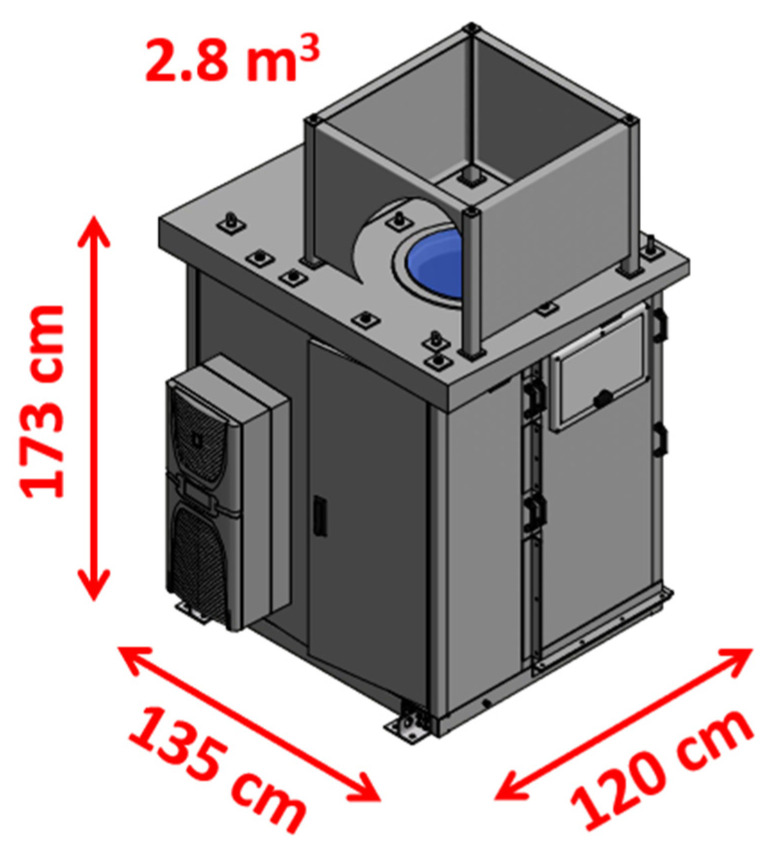
Graphical representation of the external part of the instrumental cabinet developed to host the compact Raman lidar system MARCO.

**Figure 4 sensors-23-08262-f004:**
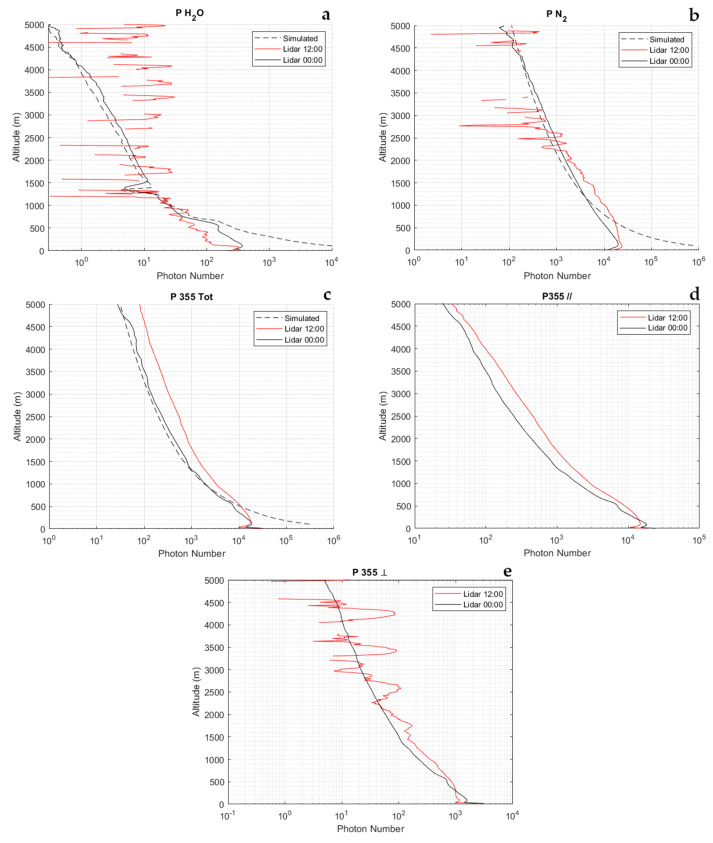
Typical amplitude of the measured signals, expressed as signals’ mean photon counts. Illustrated signals are the water vapor (**a**) and molecular nitrogen (**b**) vibrational Raman signals, $P_{H_{2}O}\left(z\right)$, and $P_{N_{2}}\left(z\right)$, respectively, and the total (**c**), the parallel (**d**), and cross-polarized (**e**) elastic signal at 355 nm, $P_{355}\left(z\right)$, $P_{355}^{\|}\left(z\right)$, and $P_{355}^{┴}\left(z\right)$, respectively. The figure also includes the corresponding signals simulated through an end-to-end performance model developed at Università della Basilicata.

**Figure 5 sensors-23-08262-f005:**
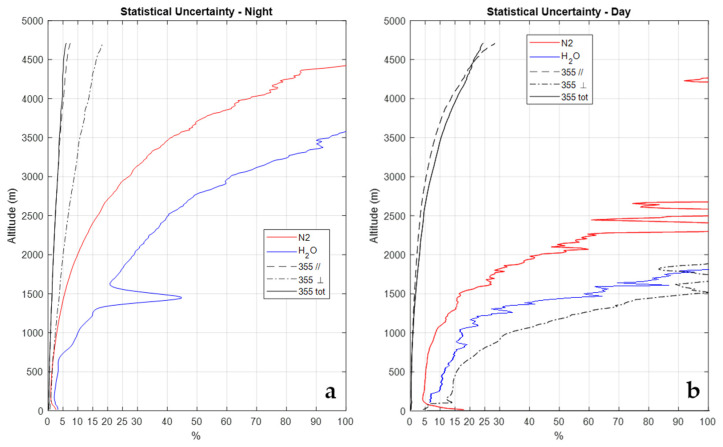
Error profiles affecting the signals illustrated in Figure 4, as obtained through the application of Poisson statistics to signal photon counts. (**a**) Night-time, (**b**) daytime.

**Figure 6 sensors-23-08262-f006:**
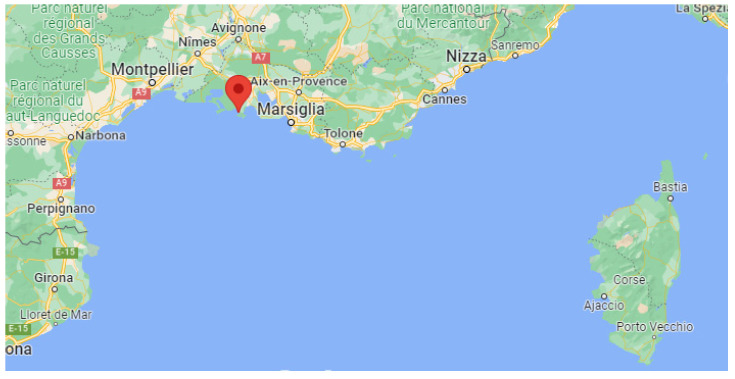
Location of the Raman lidar system MARCO operated in the frame of the Project WaLiNeAs (© Google Maps 2022).

**Figure 7 sensors-23-08262-f007:**
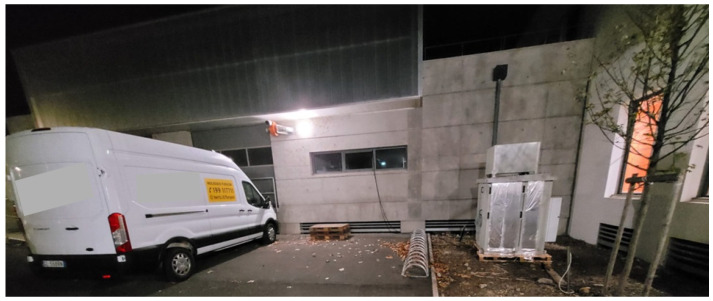
External view of the Raman lidar system MARCO (on the **right** portion of the figure) during its deployment in Port-Saint-Louis, together with the van used for its transportation (on the **left** portion of the figure).

**Figure 8 sensors-23-08262-f008:**
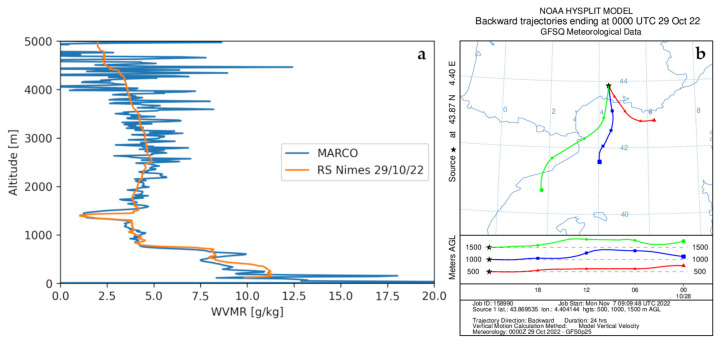
(**a**) Vertical profiles of water vapor mixing ratio $\chi_{H_{2}O}\left(z\right)$ as measured by MARCO, averaged over the 30 min time interval 21:00–21:30 UTC on 28 October 2022, and by the radiosonde launched from Nimes at 00:00 UTC on 29 October 2022. (**b**) Back-trajectory analysis started in Nimes at 00:00 UTC on 29 October 2022, considering the air masses at altitudes of 500, 1000, and 1500 m.

**Figure 9 sensors-23-08262-f009:**
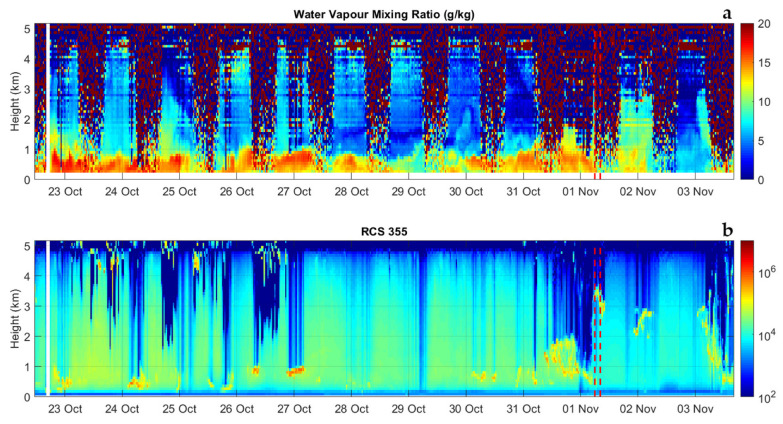
Time–height cross section of the evolution of the water vapor mixing ratio, $\chi_{H_{2}O}\left(z\right)$ (**a**) and range-corrected elastic signal at 355 nm, RCS355 (**b**) over a 12-day period from 11:30 on 22 October 2022 until 15:00 on 03 November 2022. The figure is obtained as a sequence of 587 consecutive profiles, each integrated over 30 min. The red dashed vertical lines identify the times of the radar reflectivity maps in Figure 10, showing evidence of precipitations.

**Figure 10 sensors-23-08262-f010:**
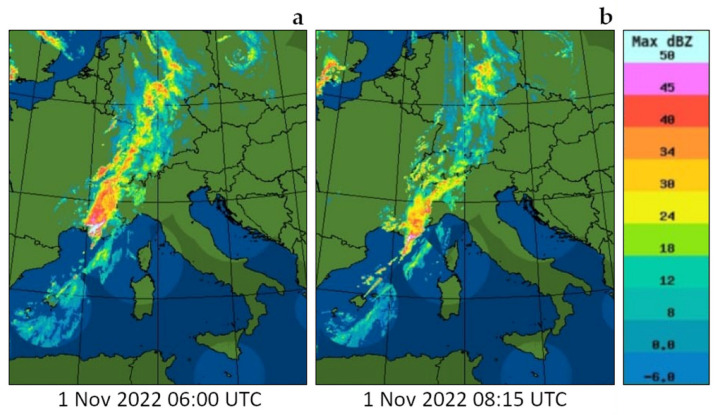
Radar reflectivity, expressed in dBZ, at 06:00 UTC (**a**) and 08:15 UTC (**b**) on 01 November 2023. Observed values, in the range 30–50 dBZ, are found to correspond to a precipitation rate of 2–3 mm per hour.

**Figure 11 sensors-23-08262-f011:**
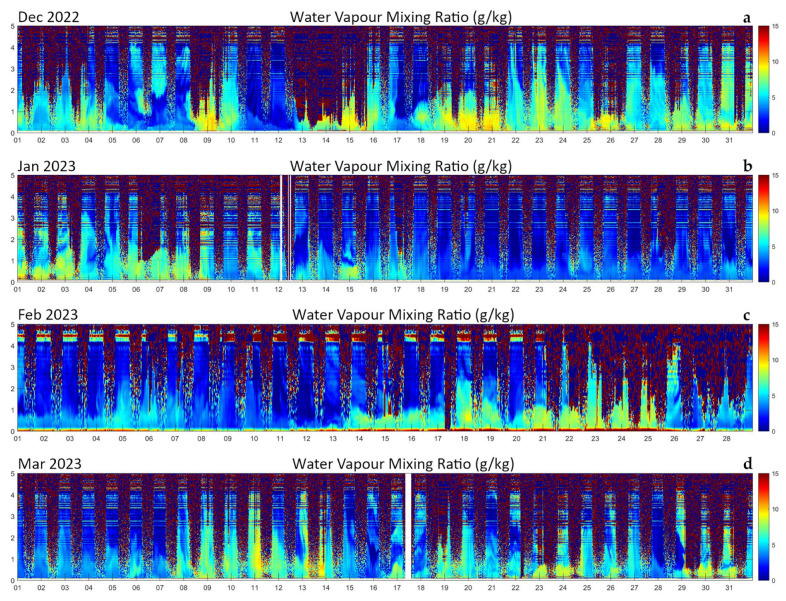
Time–height cross section of the evolution of the water vapor mixing ratio over four consecutive months (December 2022 through March 2023). (**a**) December 2022, (**b**) January 2023, (**c**) February 2023, (**d**) March 2023.

**Table 1 sensors-23-08262-t001:** Technical specification of the selected micro-pulse laser source.

Laser	Nd:YAG
Wavelength (nm)	354.7 ± 1.0
Operating mode	Acousto-Optic Q-switched
Average power (W)	1–5 (6 W @ 30 kHz)
Rep. rate (kHz)	10–40
Pulse energy @10 kHz (μJ)	190
Pulse energy @20 kHz (μJ)	250
Pulse energy @30 kHz (μJ)	200
Pulse energy @40 kHz (μJ)	140
Pulse width (ns)	~12 @ 30 kHz
Warm-up time (minutes)	<10
Transverse mode	TEM00
Beam diameter (after 10× beam expander, mm)	8
Beam divergence, full angle (after 10 × beam expander, mrad)	<0.3
Beam quality (M2)	<1.3
Polarization ratio/Direction	>100:1, Horizontal
Cooling method	Water Cooled
Oper. envir. temp. (°C)	15–35
Supply Voltage	DC 12 V (≥25 A)
Size (mm)	450 × 180 × 95
Weight (kg)	10

**Table 2 sensors-23-08262-t002:** Technical specification of the lidar receiver. CW stands for filter center wavelength, BW stands for filter bandwidths.

Telescope	Ritchey–Chrétien			
Focal length (mm)	3250			
Primary mirror dia. (mm)	406			
F number	f/8			
Secondary mirror obs. dia. (mm)	190			
Reception channels:	CW	BW	Peak Trans. (%)	Blocking
Elastic total (nm)	354.71	0.5	≥70	≥OD7
Elastic par. polar. (nm)	354.71	0.5	≥70	≥OD7
Elastic perp. polar. (nm)	354.71	0.5	≥70	≥OD7
N_2_ roto-vib. Raman (nm)	386.69	0.3	≥75	≥OD8
H_2_O roto-vib. Ram. (nm)	407.50	0.2	≥70	≥OD6
Field of view (FWHM, mrad)	0.31			
Full overlap (m)	∼500			
Detectors:	Photomultiplier tubes			
Quant. eff. @ 355 nm (%)	43			
Gain	2–4 × 10^6^			
Size (mm)	22 × 22 × 50			
Weight (g)	80			
Vertical sampling (m)	15 (analogue and photon counting)			
Vertical resolution (m)	15–100			
Time resolution (s)	3–60			
Acquisition system	16 Bit, 40 MHz analogue acquisition, 800 MHz photon counting			

## Data Availability

Data are uploaded in real time to the MeteoFrance server (link: ftp.cnrm-game-meteo.fr) and are available from the authors on request. Authors are in the process to upload the data on the publicly accessible AERIS repository (https://www.aeris-data.fr/).

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
