# Peer review of "Atmospheric Thermodynamic Profiling through the Use of a Micro-Pulse Raman Lidar System: Introducing the Compact Raman Lidar MARCO"

_sensors, 2023, doi:10.3390/s23198262_

Round 1
Reviewer 1 Report
The work described in this paper is demonstrating the feasibility of these measurements and at illustrating and discussing the high achievable performance level, with a specific focus on water vapour profile measurements. The technical solutions identified in the design of the lidar system and their technological implementation within the experimental setup of the lidar prototype are also carefully illustrated and discussed. The paper illustrates the theory behind the Raman lidar technique applied to water vapour profile measurements and the simulations carried out to assess the expected system performance and also illustrates the system set up, with a specific focus on the technical solutions identified in the design of the system and their technological implementation.
Some considerations regarding the content of paper:
- In the introduction, it is desirable to write in more detail about the state of current researches in this area.
- It is necessary to carefully check the interpretation of the variables in the formulas.
- What does it mean data1 in Figure 4?
- There is too much free space on page 10. It might be better to move the text before Figure 4.
- The Figure 8, 9 are placed outside the text of the page.
- There are only several references after 2018 year. What about current research during these five years?
- The references are provided without requirements for their design. Some of them without year of publication.
Authors should carefully examine and correct syntactic errors.
Reviewer 2 Report
This is an interesting paper and the work reported proved the very interesting results one can obtain with Raman LiDAR on the water vapour atmospheric thermodynamic profiling. The work is sound and the conclusion well substantiated. I would just suggest authors to consider works published by other groups (there is a very large, yet understandable, number of self referencing) includuing older paper (for instance from work done at the University of Florence, Italy, in particular if the profiling of components other than water vapour - and there are interesting prospects in other directions that maybe authors would like to explore in the future...- is to be considered).
Please check several formating problems (page 12 for instance)
Minor revision advised.
Reviewer 3 Report
In this manuscript, the authors present an innovative Raman LIDAR system utilizing a micro-pulsed UV laser. This system is used to measure atmospheric water vapor ratios up to 5km. The authors carefully analyze the experimental results and the associated uncertainty under various ambient conditions, and compare them with simulation results. In addition to these analyses, the authors review the theory of Raman LIDAR and present the breakdown of their system.
The system introduced in this study exhibits a remarkable level of novelty, and the results it yields hold profound scientific significance. I particularly appreciate the amount of details provided by the authors. Therefore, I recommend the publication of this manuscript. My suggestions for potential improvements are as follow.
1) In this system, high-power UV laser is used, which could be highly hazardous, particularly for human eyes. I recommend the authors comment on the safety concerns.
2) The authors mention that SNR is low during daytime. Could time-gating techniques be implemented in these situations?
3) Is this system also potentially useful for remote sensing?
